# TopEx: Topic-based Explanations for Model Comparison

**Shreya Havaldar, Adam Stein, Eric Wong & Lyle Ungar**
University of Pennsylvania, Department of Computer Science
`{shreyah,steinad,exwong,ungar}@seas.upenn.edu`

## Abstract

Meaningfully comparing language models is challenging with current explanation methods. Current explanations are overwhelming for humans due to large vocabularies or incomparable across models. We present TopEx, an explanation method that enables a level playing field for comparing language models via model-agnostic topics. We demonstrate how TopEx can identify similarities and differences between DistilRoBERTa and GPT-2 on a variety of NLP tasks.

## 1 How do we compare language models?

Language models (LMs) often exhibit differences in behavior even when trained on the same dataset. Architecture, pre-training, and hyperparameter choices can all lead to varying behaviors in the LM.

However, understanding these differences beyond comparing performance metrics is challenging. Existing post-hoc interpretability approaches primarily focus on explaining individual models as opposed to comparing models. These explanations can be generally categorized by how behavior is explained: (a) feature-based, using feature attributions (Shapley et al., 1953; Sundararajan et al., 2017); (b) example-based, using previously observed samples or generated counterfactuals; or (c) concept-based, using concepts extracted from a model's latent space (Madsen et al., 2022). Methods that fall under (b) and (c) cannot be used for comparison, as the examples and concepts are model-specific and not easily comparable across models. Methods under (a) can be used to compare models, but the tens of thousands of unique tokens renders such comparisons uninterpretable.

In order to meaningfully explain and compare LMs, we propose TopEx — a topic-based explanation method. TopEx condenses feature attributions into a model-independent explanation using topic modeling, a popular statistical method that assigns words to meaningful categories.

## 2 Topic-based Explanations (TopEx)

In this section, we outline our approach for generating topic-based explanations, which consists of two main steps: (1) calculation of feature-based scores followed by (2) aggregation into topics. Given two LMs trained on the same dataset, we first generate word-level importance scores for each model. We extract Shapley values[1] (Lundberg & Lee, 2017) for all instances and aggregate these scores into global importance scores $g_w$ for each word $w$. We then map these word-level importance scores $g_w$ to topic-level importance scores $G_t$ as follows:

$$G_t = \sum\nolimits_{w \in \text{topic}_t} P(\text{topic}_t|w)g_w \tag{1}$$

Specifically, for all words in a given topic $t$, we sum over word importance scores, weighted by word membership in each topic $P(\text{topic}_t|w)$.[2] Here, the weight comes from the topic model, which could come from an existing topic lexicon such as LIWC Pennebaker et al. (2001) or be automatically learned with e.g. Latent Dirochlet Allocation (LDA) (Blei et al., 2003). Details on token-to-word aggregation and topic weighting schemes are in Appendix B and C respectively.

---

[1]Note that our approach works with any feature-based explanation.

[2]When a word in our vocabulary is not in any topics, (e.g. punctuation, LDA stopwords or words not in LIWC) we naively treat it as a different topic. We leave other approaches, such as clustering, for future work.

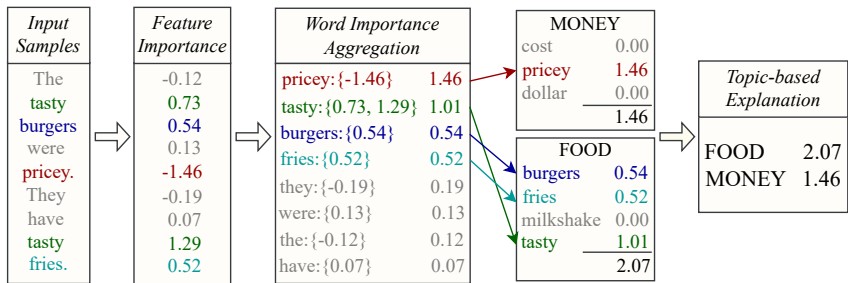

Figure 1: Generating a global explanation via TopEx. We extract an importance score for each token using Shapley values, aggregate to average global word importances, and map these importance scores to the corresponding topics for each word.

Figure 1 demonstrates our method on an example sentence. The first step of TopEx computes word-level importance scores. For example, the word "tasty" gets a an aggregate score of $1.01$ as an average of its feature attribution scores $0.73$ and $1.29$. The second step of TopEx computes topic-level importance scores. For example, scores for food-related words such as "burgers" and "fries" are aggregated with "tasty" to get the final "food" topic score. The resulting topic-based explanation is a concise summary of the model that can be used to directly compare with other models.

## 3 TopEx Explains Differences Between Models

We compare fine-tuned DistilRoBERTa (Sanh et al., 2019) and GPT-2(Radford et al., 2019) on the Yelp Reviews dataset (Zhang et al., 2015) and the GoEmotions dataset (Demszky et al., 2020). From our topic-based explanations of these models, $G^{\text{BERT}}$ and $G^{\text{GPT}}$, we calculate the distance between explanations as $G^{\Delta} = \|G^{\text{BERT}} - G^{\text{GPT}}\|_1$. The two topics with the most different and most similar importance scores are highlighted in Figure 2. We can see that DistilRoBERTa focuses more than GPT-2 on descriptions of dining when classifying a 5-star rating, while GPT-2 looks more at negativity than DistilRoBERTa. In this case, GPT-2 may be determining bad reviews through negative words, while DistilRoBERTa has learned to better recognize descriptions of dining experiences characteristic of 5-star reviews. Experiment details and additional results are given in Appendix D and E.

**Conclusion.** The vast array of possible LM architectures and training schemes motivates the need to deeper understand differences in model behavior beyond performance metrics. In this work, we present TopEx, a method that enables direct model comparisons via model-agnostic topics that can reveal *why* and *how* models behave differently.

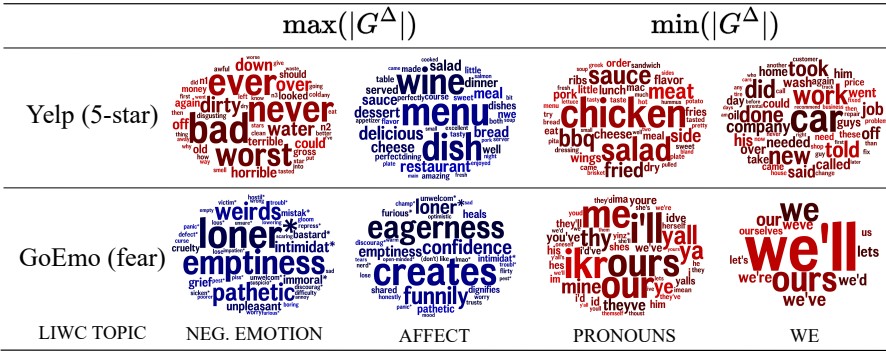

Figure 2: We explain differences in behavior of DistilRoBERTa and GPT-2 via $G^{\Delta}$. We show the two topics with most different ($\max(|G^{\Delta}|)$) and most similar ($\min(|G^{\Delta}|)$) importances between models, using LDA topics for Yelp and LIWC topics for GoEmotions. Topic visualizations in blue indicate $G_t^{\Delta} > 0$ (i.e. the topic is more important for DistilRoBERTa), while red indicates $G_t^{\Delta} < 0$ (i.e. the topic is more important to GPT-2).

URM STATEMENT

The authors acknowledge that the first and second authors of this work meet the URM criteria of ICLR 2023 Tiny Papers Track.

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

## A  APPENDIX

We include additional information detailing TopEx. Appendix B explains token-level to word-level importance score aggregation, Appendix C details topic membership weighting, Appendix D provides experiment details, and Appendix E shows additional results and topic visualizations.

## B  TOKEN-TO-WORD AGGREGATION

This section details the aggregation from token-level Shapley values, $[v_1^i, v_2^i, \ldots, v_{K_i}^i]$ to global word-level importance scores, $[g_1, g_2, \ldots, g_V]$, where $K_i$ is the total number of tokens in the $i$th input and $V$ is the size of our vocabulary.

For each of our models, we extract Shapley values, $v^i = [v_1^i, v_2^i, \ldots, v_{K_i}^i]$, computed with Partition SHAP (Lundberg & Lee, 2017) for each token $x_k^i$ in the $i$th input $s^i = [x_1^i, x_2^i, \ldots, x_{K_i}^i]$. We then calculate the Shapley values for each word $\hat{x}_w^i$ by summing over its constituent tokens. We write the $i$th word-level input as $\hat{s}^i = [\hat{x}_1^i, \ldots, \hat{x}_{W_i}^i]$ with corresponding word-level Shapley values $\hat{v}^i = [\hat{v}_1^i, \ldots, \hat{v}_{W_i}^i]$ for the $W_i$ words in the $i$th input. This calculation of local word-level Shapley values is shown in Equation 2.

From local word-level Shapley values, we derive global word-level importance scores by aggregating the absolute value of local word-level Shapley values over each word as shown in Equation 3. Note that we choose to take the absolute value to aggregate over magnitude of importance.

$$\hat{v}_j^i = \sum_{x_k \in \hat{x}_j} v_k^i \quad (2) \qquad\qquad g_w = C(w) \sum_{i=1}^{N} \sum_{j=1}^{|\hat{s}^i|} \mathbb{1}_{[\hat{x}_j = w]} |\hat{v}_j^i| \quad (3)$$

The weighting $C(w)$ in Equation 3 can be set to balance between the impact of word frequency and word importance when aggregating local to global explanations. One choice for $C(w)$ is simply $C(w) = 1$, which is the traditional way of aggregating local explanations through summation.

An alternative is the inverse of the number of times a word appears in the dataset,

$$C(w) = \frac{1}{\sum_{i=1}^{N} \sum_{j=1}^{|\hat{s}^i|} \mathbb{1}_{[\hat{x}_j = w]}}, \quad (4)$$

which removes the effect of word frequency from the global word importance. Results in Appendix E use the above weighting and thus do not take into account word frequency when mapping to word-level importance scores.

### B.1  PRESERVING ADDITIVITY

Note that we can slightly modify the above equations to first compute local topic-level importance scores that preserve Shapley additivity, and then aggregate from local to global topic-based explanations, generating a faithful explanation.

Equation 5 describes aggregation from local word-level Shapley values $\hat{v}^i = [\hat{v}_1^i, \ldots, \hat{v}_{W_i}^i]$ to a local word-level importance score, $l_w^i$ for word $w$ in the $i$th input. We then aggregate from the local word-level importance score to a local topic-level importance score, $l_t^i$ for the $t$th topic, shown in Equation 6. Lastly, Equation 7 details how to aggregate local topic-level importance scores, $L^i = [L_1^i, \ldots, L_T^i]$ to compute a global topic-level explanation.

$$l_w^i = \sum_{j=1}^{|\hat{s}^i|} \mathbb{1}_{[\hat{x}_j = w]} \hat{v}_j^i \quad (5) \qquad L_t^i = \sum_{w \in \text{topic}_t} P(\text{topic}_t | w) l_w^i \quad (6) \qquad G_t = \sum_{i=1}^{N} |L_t^i| \quad (7)$$

This modified aggregation provides a way to compute a topic-based explanation for a single instance that preserves the additive property of Shapley values, as local topic-based explanations for an instance will sum to the model's output for that instance. Additionally, the sum of the topic importances equals the total effect.

## C    Topic Membership Weighting

We describe the value of $P(\text{topic}_t|w)$ in Equation 1 and 6 for various topic modeling methods. When using LDA topics, this weight is learned by the topic model, and comes from $P(w|\text{topic}_t)$. To derive a topic distribution, $P(\text{topic}_t|w)$, from the topic model's word distibution, $P(w|\text{topic}_t)$, for each topic, we simply renormalize the word distributions:

$$P(\text{topic}_t|w) = \frac{P(w|\text{topic}_t)P(\text{topic}_t)}{P(w)} \propto \frac{P(w|\text{topic}_t)}{\sum_w P(w|\text{topic}_t)}. \tag{8}$$

This works under the assumption that $P(\text{topic}_t)$ is equal for all topics, which holds for LDA due to the use of the symmetric Dirichlet distribution.

For LIWC and other unweighted topic lexicons, this membership weight is $1/T_w$, where $T_w$ is the number of topics a word appears in.

## D    Experiment Details

**Datasets**    We fine-tune DistilRoBERTa and GPT-2 on three classification tasks: the Yelp Reviews dataset (Zhang et al., 2015), a polarity detection task based on the number of stars associated with a text review; the GoEmotions dataset (Demszky et al., 2020), an emotion classification task where we scope to only the six Ekman emotions; and the Blog Authorship Corpus (Schler et al., 2006), an authorship attribution task where we scope to age and gender[3].

**Models**    Both models were trained with Adam using 3 epochs, a learning rate of $5.00\text{e}-5$, and half precision. DistilRoBERTa was trained with a batch size of 64 and GPT-2 was trained with a batch size of 32. The maximum token length was always set to 512 so that both DistilRoBERTa and GPT-2 were trained on the same data. For GoEmotions, we add an output layer of size 6 use binary cross entropy loss for multilabel classification. Similarly, for Blog Authorship we use binary cross entropy loss and an output layer of size 5 (for two genders and 3 age groups). For Yelp we use an output layer of size 5 and train using cross entropy loss for single label classification into 1-5 stars.

Table 1 shows the resulting accuracies and F1 scores for these tasks. For multilabel classification datasets (Blog and GoEmotions), the table contains the average accuracy and F1 score across all classes. For Yelp Reviews, we combine 1-2 star classifications (negative reviews) and 3-4 star classifications (positive reviews) and report performance metrics on polarity classification.

**Topic Modeling**    To get our LDA topics, we run MALLET (Graham et al., 2012) with 30 topics, $\alpha = 5.0$, and $\beta = 0.01$. We use the 100 most frequent words in each dataset as stopwords. We use the standard LIWC lexicon (Pennebaker et al., 2001) and treat each category as a unique topic with identical weighting for words within each category.

## E    Full Results

For models trained on Yelp Reviews and the Blog Authorship Corpus, we use TopEx with Partition SHAP Lundberg & Lee (2017) for feature attribution and LDA for topic modeling. For models trained on GoEmotions, we use TopEx with Partition SHAP for feature attribution and LIWC topics.

Table 2 shows the three most important topics and least important topics for each model, along with the corresponding topic importance score. These scores are L1 normalized for direct numerical comparison between models. We also calculate $||G^{\text{BERT}}||_1 - ||G^{\text{GPT}}||_1 = G^\Delta$ to measure topic-wise differences in importance. The topics with the greatest magnitude in this residual explanation are shown in the rightmost column of Table 2. Specifically, we show the three topics with the most different importance scores $(\max(|G^\Delta|))$ and the most similar importance scores $(\min(|G^\Delta|))$ between models.

---

[3]Gender was measured by a binary label, and we note this does not cover the entire population of possible bloggers.

Table 1: Accuracy and F1 score for trained models on the three benchmark datasets. For multilabel classification datasets (Blog and GoEmotions), we report the average accuracy and F1 score across all classes as well as per class metrics. For the Yelp dataset, we report the accuracy on the standard polarity task as well as accuracy for predicting 5-star reviews.

| | DistilRoBERTa | | GPT-2 | |
|---|---|---|---|---|
| | F1 | Accuracy | F1 | Accuracy |
| Blog (Avg.) | 69.2 | 46.9 | 69.0 | 45.9 |
| Blog (Female) | 69.2 | 69.4 | 70.3 | 69.8 |
| Blog (23-33) | 72.6 | 72.7 | 71.4 | 72.1 |
| Yelp (Polarity) | 93.3 | 91.9 | 92.5 | 90.9 |
| Yelp (5-star) | 76.2 | 78.1 | 74.1 | 76.2 |
| GoEmotions (Avg.) | 53.5 | 87.2 | 58.7 | 88.0 |
| GoEmotions (Joy) | 60.4 | 97.7 | 62.3 | 97.9 |
| GoEmotions (Fear) | 69.8 | 99.1 | 68.4 | 99.1 |

Table 2: TopEx on three benchmark datasets. We do a manual evaluation of LDA results to name topics, and show further topic visualizations Appendix E.1

| | $G^{\text{BERT}}$ | | $G^{\text{GPT}}$ | | $G^{\text{BERT}} - G^{\text{GPT}}$ | |
|---|---|---|---|---|---|---|
| | Topic | Importance | Topic | Importance | Topic | Importance |
| Blog (Female) | technology | 4.82e−2 | technology | 4.46e−2 | animals | −4.91e−3 |
| | texting | 4.20e−2 | animals | 4.35e−2 | technology | 3.57e−3 |
| | travel | 3.90e−2 | texting | 4.29e−2 | "xbubzx" | −3.31e−3 |
| | descriptors | 1.02e−2 | descriptors | 9.33e−3 | games | −7.79e−6 |
| | longing | 1.07e−2 | longing | 1.20e−2 | school/work | 1.28e−4 |
| | temporal | 1.48e−2 | temporal | 1.46e−2 | temporal | 2.04e−4 |
| Blog (23-33) | technology | 5.59e−2 | technology | 5.01e−2 | texting | −6.54e−3 |
| | animals | 4.06e−2 | animals | 4.54e−2 | technology | 5.87e−3 |
| | books/movies | 4.01e−2 | texting | 4.26e−2 | animals | −4.87e−3 |
| | descriptors | 1.01e−2 | descriptors | 8.95e−3 | politics | −4.17e−5 |
| | longing | 1.08e−2 | longing | 1.20e−2 | music | −2.85e−4 |
| | temporal | 1.41e−2 | temporal | 1.30e−2 | economy | 3.20e−4 |
| Yelp (5-star) | atmosphere | 6.51e−2 | review | 6.18e−2 | dining | 1.21e−2 |
| | review | 5.60e−2 | atmosphere | 6.10e−2 | negativity | −1.19e−2 |
| | entertainment | 5.55e−2 | entertainment | 4.74e−2 | conversation | −1.17e−2 |
| | time | 1.31e−2 | visiting | 1.75e−2 | american food | −5.14e−5 |
| | visiting | 1.50e−2 | time | 1.81e−2 | labor | −4.84e−4 |
| | location | 2.03e−2 | breakfast | 2.20e−2 | french | −5.82e−4 |
| GoEmo (Fear) | AFFECT | 0.172 | AFFECT | 8.92e−2 | NEGEMO | 9.01e−2 |
| | NEGEMO | 0.161 | NEGEMO | 7.11e−2 | AFFECT | 8.27e−2 |
| | ANX | 0.102 | ADJ | 4.84e−2 | ANX | 6.32e−2 |
| | FILLER | 1.57e−5 | WE | 1.11e−4 | PPRON | −1.10e−5 |
| | WE | 3.93e−5 | FILLER | 1.43e−4 | WE | −7.21e−5 |
| | YOU | 5.31e−5 | YOU | 2.56e−4 | SHEHE | −1.10e−4 |
| GoEmo (Joy) | AFFECT | 0.174 | AFFECT | 0.122 | POSEMO | 5.38e−2 |
| | POSEMO | 0.161 | POSEMO | 0.107 | AFFECT | 5.16e−2 |
| | ADJ | 6.38e−2 | ADJ | 4.59e−2 | ADJ | 1.80e−2 |
| | WE | 8.89e−5 | WE | 6.97e−5 | WE | 1.92e−5 |
| | SHEHE | 1.68e−4 | THEY | 2.04e−4 | ARTICLE | 1.96e−5 |
| | YOU | 1.99e−4 | FILLER | 2.33e−4 | INGEST | −3.17e−5 |

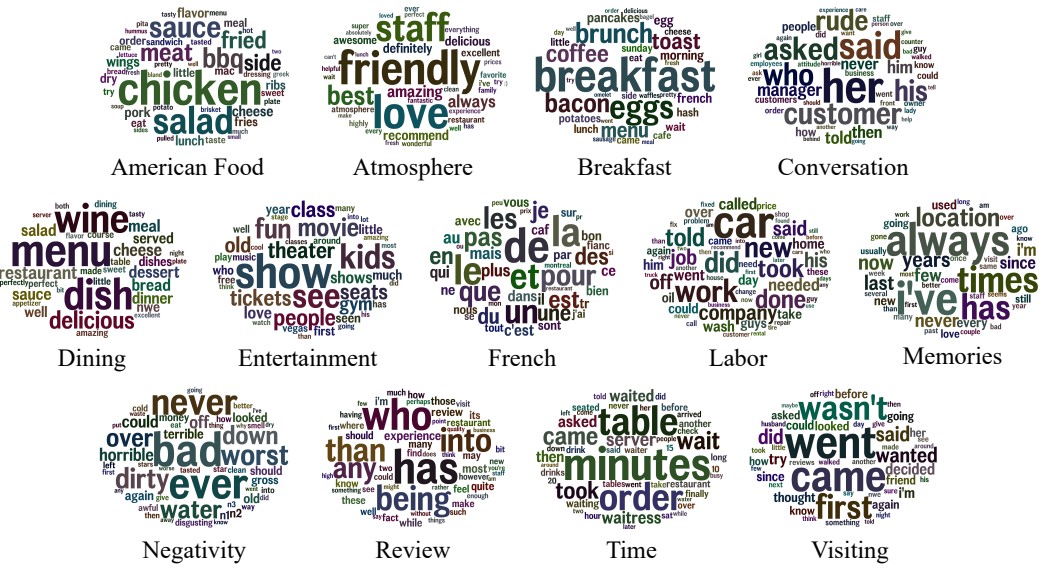

Figure 3: Visualizations for LDA topics on the Yelp Reviews dataset

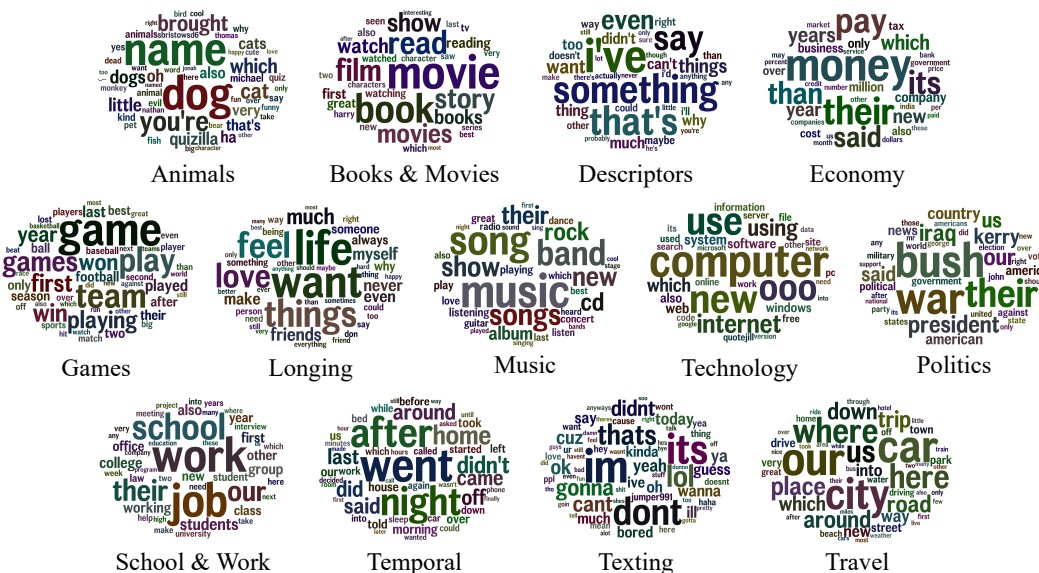

Figure 4: Visualizations for LDA topics on the Blog Authorship Corpus

## E.1 TOPIC VISUALIZATIONS

We show word clouds to visualize all LDA topics shown in Table 2. Topics are named based on manual evaluation of the top 15-20 words within each topic. We find all topics had some unifying theme and were easy to name. Figure 3 contains the Yelp Reviews topics and Figure 4 contains the Blog Authorship Corpus topics.

