# OpenReview forum: "TopEx: Topic-based Explanations for Model Comparison"
_ICLR.cc/2023/TinyPapers — Submitted to Tiny Papers @ ICLR 2023_

### Official Review · Reviewer_Whgs · 2023-03-20

**Confidence:** 3

**Summary Of Contributions:**

An interesting paper for model comparsion

**Rating:**

Clear, Correct, and Reproducible (CCR): a submission which meets the reviewing criteria

**Strengths And Weaknesses:**

This paper introduces TopEx, an explanation method that facilitates fair comparison of language models by using model-agnostic topics. The paper demonstrates how TopEx can identify similarities and differences between DistilRoBERTa and GPT-2 across a range of NLP tasks.

Strengths:

1.It is an interesting approach to use model-agnostic topics for model comparison.

2.The paper includes several pivot experiments.

Weaknesses:

1.It is not easy to evaluate the impact of the model comparison on downstream tasks or model interpretability.

2.The technical novelty is limited, and the paper could benefit from the use of stronger topic models.

**Suggested Changes:**

1.It is not easy to evaluate the impact of the model comparison on downstream tasks or model interpretability.

2.The technical novelty is limited, and the paper could benefit from the use of stronger topic models.

---

### Official Review · Reviewer_PR9E · 2023-03-28

**Confidence:** 4

**Summary Of Contributions:**

This paper proposes a novel way of using topic modeling to compare the preferences of different pretrained models

**Rating:**

High Potential (HP): a submission which meets the reviewing criteria and has potential to make an impact on the field

**Strengths And Weaknesses:**

This paper proposes TopEx, a method to condense feature attributions into a model-independent explanation using topic modeling. The approach is to identify important words in the context and checking the topics where these words belong to. By using a model-agnostic approach like this, the authors observe that different models assign different importance to various parts of the context.

It's quite a simple and effective approach to get insight into how pretrained models work. There is no automation and in order to get the insight one has to look into examples case by case. Nevertheless it's an additional qualitative analysis of these models.


**Suggested Changes:**

Question: it is not clearly explained how "word-level importance scores for each model" is computed. Is it the likelihood that the model assigns to each word at generation? How do the authors get word-level importance from subword-level importance?

---

### Author Response · Authors · 2023-06-01
**Opt-in for Archival**

We thank the reviewers for their feedback! We have incorporated suggested changes and made the description of our method more clear.
We would also like to opt-in and archive our work.

---

### Meta-Review · Area_Chair_iCUY · 2023-04-07

**Recommendation:** Invite to present
**Confidence:** 4

**Metareview:**

This is a nice, simple, and effective approach to provide explanation of model's decisions using topic modelling. The approach is to identify important words in the context and checking the topics where these words belong to. They use a model-agnostic approach like topic modelling and observe that different models assign different importance to various parts of the context. The paper is well-written and the hypothesis is supported by the experiments.
Some clarifications is needed (see reviewers' comments, for instance clarifying how do the authors get word-level importance from subword-level importance?) but overall a valuable work.


**Summary:**

A topic-based explanation method to compare LMs which is simple enough that can be an additional qualitative analysis to most comparisons.

**Comments And Feedback To The Authors:**

By addressing reviewers' comments and questions, this paper can become a good contribution to the literature.



**Reason For Not Giving A Higher Recommendation:**

Need some clarifications in writing to explain the details of the approach in a better way.

**Reason For Not Giving A Lower Recommendation:**

It's a valuable idea and execution and will be useful for the researchers in the field.

---

### Decision · Program_Chairs · 2023-04-08

Invite to present